# Healthcare financing and social protection policies for migrant workers in Malaysia

**Tharani Loganathan**[1]*, **Zhie X. Chan**[2], **Nicola S. Pocock**[2,3]

**1** Centre for Epidemiology and Evidence-based Practice, Department of Social and Preventive Medicine, University of Malaya, Kuala Lumpur, Malaysia, **2** United Nations University—International Institute for Global Health (UNU-IIGH), Kuala Lumpur, Malaysia, **3** Gender Violence & Health Centre, London School of Hygiene and Tropical Medicine, London, United Kingdom

* drtharani@ummc.edu.my

**Data Availability Statement:** All relevant data are within the manuscript and its Supporting Information files.

**Funding:** We are grateful for funding to conduct this research from the Asia Pacific Observatory on

## Abstract

### Background

For Malaysia, a nation highly dependent on migrant labour, the large non-citizen workforce presents a unique health system challenge. Although documented migrant workers are covered by mandatory healthcare insurance (SPIKPA), financial constraints remain a major barrier for non-citizen healthcare access. Malaysia recently extended protection for migrant workers under the national social security scheme (SOCSO), previously exclusive to citizens. This study aims to evaluate healthcare financing and social security policies for migrant workers to identify policy gaps and opportunities for intervention.

### Methods

A total of 37 in-depth interviews were conducted of 44 stakeholders from July 2018 to July 2019. A mixed-methods analysis combining major themes from qualitative interviews with policy document reviews was conducted. Descriptive analysis of publicly available secondary data, namely revenues collected at government healthcare facilities, was conducted to contextualise the policy review and qualitative findings.

### Results

We found that migrant workers and employers were unaware of SPIKPA enrolment and entitlements. Higher fees for non-citizens result in delayed care-seeking. While the Malaysian government nearly doubled non-citizen healthcare fees revenues from RM 104 to 182 million (USD 26 to 45 million) between 2014 to 2018, outstanding revenues tripled from RM 16 to 50 million (USD 4 to 12 million) in the same period. SPIKPA coverage is likely inadequate in providing financial risk protection to migrant workers, especially with increased non-citizens fees at public hospitals. Undocumented workers and other migrant populations excluded from SPIKPA contribution to unpaid fees revenues are unknown. Problems described with the previous Foreign Workers Compensation Scheme (FWCS), could be partially addressed by SOCSO, in theory. Nevertheless, questions remain on the feasibility of implementing elements of SOCSO, such as recurring payments to workers and next-of-kin overseas.

Health Systems and Policies (APO) [IF034-2020] and the China Medical Board's Equity Initiative [IF055-2018]. The funders had no role in study design, data collection and analysis, decision to publish, or preparation of the manuscript.

**Competing interests:** The authors have declared that no competing interests exist.

## Conclusion

Malaysia is moving towards migrant inclusion with the provision of SOCSO for documented migrant workers, but more needs to be done. Here we suggest the expansion of the SPIKPA insurance scheme to include all migrant populations, while broadening its scope towards more comprehensive coverage, including essential primary care.

## Introduction

Global migration for work is the largest driver of international migration with 164 million migrant workers estimated in 2017, accounting for nearly two-thirds of all international migrants [1, 2]. International commitment towards protecting migrant workers' rights is most recently embodied in the 2018 Global Compact for Safe, Orderly and Regular Migration [3]. Although health systems have pledged to ensure 'no one is left behind' and to achieve Universal Health Coverage with the 2030 Agenda for Sustainable Development, the health of migrant populations are often overlooked [4, 5].

Malaysia is an upper-middle income nation dependant on migrant labour, with migrant workers shouldering employment in low-skilled jobs that citizens are reluctant to perform [6, 7]. Migrant workers are employed in five major, labour-intensive sectors: manufacturing, construction, services, plantations and agriculture [8]. The Ministry of Home Affairs estimates two million documented migrant workers in Malaysia in 2019 [9]. Others estimate up to 5 million migrant workers including undocumented workers in the country, or nearly a sixth of Malaysia's population of 32 million, presenting a unique challenge to the health system [10].

Malaysia has been lauded as having achieved Universal Health Coverage with its tax-financed public healthcare system provided mainly by the Ministry of Health (MOH), Malaysia [11]. While fees at public healthcare facilities are highly subsidised for citizens, non-citizens' fees have been substantially increased, reflecting healthcare rationing [12, 13]. Despite the introduction of the Foreign Worker Hospitalization and Insurance Scheme (SPIKPA) to provide migrant workers with financial risk protection against healthcare expenditure incurred with inpatient care or surgery at MOH hospitals, financial constraints remain a considerable obstacle towards healthcare access in Malaysia [14, 15].

The provision of social security for workers in Malaysia has until recently been inherently unequal, with migrant workers covered against workplace accidents by the Workmen's Compensation Act 1952 (Act 273), while citizens receive protection under the Employees' Social Security Act, 1969 (Act 4) [16–18]. In a shift towards migrant inclusion, the Malaysian government placed migrant workers' social security protections with the national social security organisation (SOCSO) commencing January 2019.

In this paper, we examine the evolution of healthcare policy for migrant workers in Malaysia, while evaluating healthcare financing and social security policies and other options for fiscal space, in order to identify gaps and opportunities to improve migrant health financing and coverage.

## Materials and methods

### Study design

Policy document review and thematic analysis of qualitative interviews were combined with secondary quantitative data to evaluate healthcare financing and social security policies for

migrant workers. We combined major themes from interview data with policy analysis and descriptive quantitative data is presented to contextualize findings, in an integrated Results section. Our mixed methods approach provides insights at critical moments in the analysis, rather than a standardized mixed methods approach, which has been critiqued for taking an unreflective and mechanical approach to knowledge production [19].

### Definition of terms

Documented and undocumented migrant workers, commonly called 'foreign workers' in Malaysia are the population of interest in this study. Non-citizens are a wider umbrella term, which includes other migrant populations like refugees, asylum seekers, victims of trafficking and expatriates, that are not the primary focus of this study.

We define a migrant worker as a person who crosses international borders for employment. Documented or regular migrants possess legal documents such as passports and work permits and are authorised to enter, reside and partake in employment in the country. Undocumented or irregular migrants do not have the required legal documents or authorisation to enter, reside or be employed in the country officially [20, 21].

### Data collection and analysis

For the document review, Malaysian health and labour laws, policy documents, guidelines and circulars relating to healthcare delivery to migrants, and reports of local and international organisations concerning migrant health from January 2001 to July 2019 were retrieved from relevant Ministry and local organisation websites (MOH, Ministry of Human Resources, Department of Statistics, PERKESO) and international organisation websites (IOM, ILO, WHO) and reviewed.

For the qualitative component, data collection was conducted from July 2018 to July 2019. Semi-structured interview guides were developed, and questions were adapted depending on the participants' organisational backgrounds and knowledge. See S1 File for interview guides. Participants were sampled purposively using an initial sampling frame from a migrant health stakeholder workshop [22]. Further recruitment was done by participant referral and purposefully identifying stakeholders through LinkedIn. Interviews were conducted until theoretical saturation was reached.

We conducted 37 in-depth interviews of 44 individuals including those from civil society and international organisations, trade unions, academia, industry, as well as medical doctors, migrant workers and other policy stakeholders (Table 1). Most interviews were conducted on an individual basis; however, several interviews were conducted with small groups of 2 or 3 participants from the same organisation. Study participants were involved in case management, legal aid, employment, training, research or medical service provision for migrant workers. We interviewed migrant workers and their representatives from major migrant-sending countries like Indonesia, Bangladesh, Nepal and the Philippines. The medical professionals interviewed were doctors working in public and private healthcare facilities and civil society organisation (CSO) clinics providing free healthcare to non-citizens.

Interviews were conducted in English and Bahasa Malaysia (Malay language), by the research team (TL, ZC and NP). Audio recordings were transcribed verbatim. Audio recordings and electronic transcripts were stored in secure data servers, while printed transcripts and notes were stored in a locked cupboard. Data were analysed thematically in an immersive, exploratory and inductive manner [23]. Transcripts were coded into emerging themes using NVivo 12 separately by TL and ZC. Codes and themes were refined by repeated readings of transcripts and regular discussions, giving due attention to negative themes and minor quotes.

Table 1. Characteristics of the study participants (n = 44).

| Participant Background | Label | No. |
|---|---|---|
| Medical Doctor | MD | |
| Public | | 4 |
| Private | | 6 |
| Civil society organisation | | 3 |
| Civil society organisation | CSO | 10 |
| Industry | IND | 5 |
| Migrant worker[1] | MW | 4 |
| International organisation | IO | 4 |
| Trade union | TU | 3 |
| Academia | AC | 3 |
| Other policy stakeholders[2] | POL | 2 |
| **Total** | | **44** |

[1] Only 1 of the 4 migrant workers interviewed identified himself as a worker only. Others were also members of civil society organisations (2) or trade unions (1).

[2] Government or government-linked organisation.

Interviews in Bahasa Malaysia were analysed in the same language, while extracted quotations were translated for publication.

To contextualise the policy review and qualitative findings, we analysed publicly available secondary data on medical revenues collected and outstanding revenues in accordance with the Fees Act 1951, for both citizens and non-citizens at Ministry of Health healthcare facilities. This data was sourced from the Ministry of Health, Malaysia Annual Reports from 2008 to 2018 [24]. This descriptive analysis aimed to examine changes in revenues collected and outstanding revenues alongside the evolving financial policies in Malaysia. Publicly available data on other migrant-related charges and taxes, including annual levies which employers are required to pay for the employment of migrant workers, were also descriptively analysed. All costs are reported in Malaysian Ringgit (RM) and United States Dollars (USD), using the 2018 World Bank exchange rate of 4.04 [25].

## Ethics

Participant information sheets were distributed, and informed consent was obtained at recruitment. All participants agreed to be audio recorded and quoted anonymously in publications. Participants were informed that study participation was voluntary, and they would at any point, be able to refuse to answer questions or terminate the interview. Interviews were primarily conducted by a medical doctor (TL) and academic researchers (ZC, NP) respectively. Interviewers were likely to be viewed as trusted authority figures, particularly with migrant workers. As described elsewhere [14], interviews were conducted at locations and times of study participants choice, to minimise the effects of social position and power imbalances. Migrant participants, in particular, were assured that they could refuse to answer questions or to end the interview at any time. In doing so, we hoped that participants felt that they could exert a degree of control over the interview process [14].

Ethical approval to conduct this study was obtained from the Medical Ethics Committee, University Malaya Medical Centre and the Medical Research and Ethics Committee, Ministry of Health, Malaysia (Approval numbers: UM.TNC2/UMREC-238 and NMRR-18-1309-42043).

## Results

### Evolution of migrant healthcare and social security policies in Malaysia

Healthcare policy for migrants in Malaysia has unfolded rapidly over the past two decades but remains centred primarily on security and sovereignty. See Fig 1 for the evolution of major migrant healthcare and social security policies in Malaysia from 2001 to the present day.

In 2001, the offices of the Director General of Health released guidelines for the reporting of undocumented migrants seeking care at MOH clinics and hospitals [28]. Health workers were reminded of their duty as civil servants to report undocumented migrants including children, migrant workers, refugees and asylum seekers, to the police per the Immigration Act 1959/1963 (amend 1997) [29]. The circular explained the rationale for this policy in two ways. First, non-citizens were framed as a potential national security threat requiring collective action. Second, non-citizens were seen as taking up a large portion of the MOH budget meant for Malaysian citizens [15].

Keeping with the justification of scarcity of healthcare resources and rationing of services delivered to non-citizens, the Malaysian government began phasing out subsidised healthcare for non-citizens by imposing increased medical fees to non-citizens with the enforcement of Fees (Medical) (Cost of Services) Order 2014 (S1 Table. Charges for Malaysian citizens and non-citizens at public clinics and hospitals) [12, 14, 30]. Initially, the fee increase was to be implemented incrementally over four years, starting in January 2015. However, full non-citizen fees were enforced from January 2016, two-years ahead of the initial target of 2018, without explanation on reasons for the change in implementation [31, 32]. Furthermore, except for certain exemptions, medication prescribed to non-citizens for the treatment of non-communicable diseases would only be supplied for five days at public facilities. A notable exemption to this policy is for the treatment of seven infectious diseases, justified as a measure to protect Malaysian citizens from the threat of communicable disease among migrants [33].

In a somewhat unprecedented move, the Malaysian government announced that the Foreign Workers Compensation Scheme (FWCS), which primarily addressed accident compensation and repatriation in case of death, would be phased out in favour of migrant inclusion in the Social Security Scheme (SOCSO) from January 2019, on near parity terms with Malaysian citizens. SOCSO includes health provision for occupational injuries and disease, including free treatment at SOCSO panel clinics and government hospitals [27].

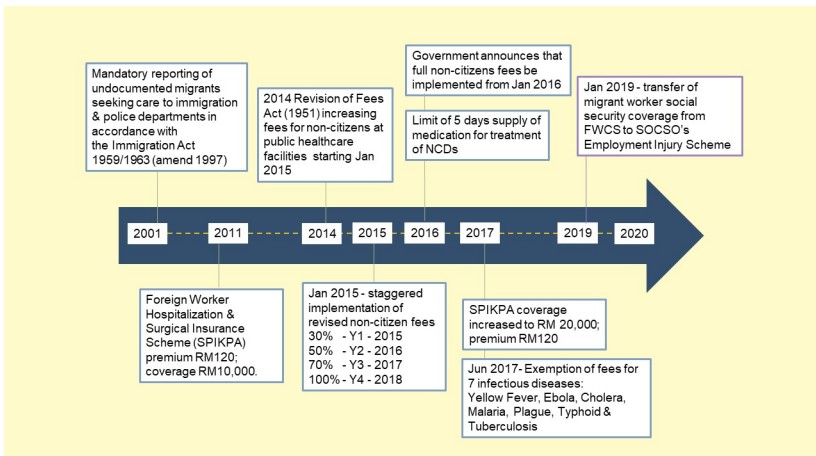

**Fig 1. Timeline of the evolution of healthcare and social security policies for migrant workers in Malaysia.**
Sourced from [26, 27].

Study participants identified several challenges and policy gaps for migrant workers seeking care in Malaysia related to the evolving legal and policy framework since 2001, relevant for healthcare financing and social security (Table 3). We describe the major challenges and gaps in healthcare financing (SPIKPA) and social security policies (FWCS and SOCSO) in the next sections. Health protection schemes for healthcare financing and social security available for migrant workers in Malaysia are detailed in Table 2. A summary of major themes and sub-themes in this section is detailed in Table 3.

**Table 2. Health protection schemes for foreign workers in Malaysia.**

| Insurance/ Protection Scheme | Established | Provision | Basic Mechanism | Strengths | Weaknesses |
|---|---|---|---|---|---|
| SPIKPA: Foreign Workers' Insurance Protection Scheme | January 2011 | Immigration (Department of Labour, MOHA) & Health Policy (MOH)* | • Private Insurance from 25 providers | • Cashless | • Low awareness of entitlements |
| | | | • Covers hospitalisation and surgical charges at Public Hospitals (MOH) | • No deposit required upon checking into the hospital (E-System) | • Does not cover outpatient services |
| | | | • RM 120 (USD 30) per annum; paid by the worker | • Designed to reduce the financial burden of the employers | • Does not cover plantation and domestic workers |
| | | | • Annual limit of up to RM 10,000 (RM 20,000 from end 2016) (increased from USD 2500 to USD 5000 in 2016) | | • Although there was an increase in annual limit, this is insufficient for management of severe cases or those requiring long-term care |
| FWCS: Foreign Worker Compensation Scheme | 1998–2019** | Section 26 (2) of Workmen's Compensation Act 1952 (Amended Aug 1996) | • 26 private insurers | • 'No fault' compensation | • Relies on the employer to make the claims |
| | | | • RM 72 (USD 18) per annum; paid by the employer, without any salary deduction | • Covers injuries, occupation diseases and fatalities related to employment | • Claim process can take time and delay access to the funds required for emergencies |
| | | | • Lump-sum compensation, no more than RM 23,000 (USD 5,700) for injuries and RM 25,000 (USD 6,196) for death | • It is an offence for employers to deduct FWCS premium from workers' salary | • If employers advanced payment of medical bills, then the amount will be deducted from compensation received |
| | | | • Labour Commissioner will assess all compensation payable | | • |
| SOCSO: Social Security Organisation | 1971-1993/ January 2019*** | Employees' Social Security Act 1969 (Act 4) | • Employment Injury Scheme (EIS) | • 'No fault' compensation | • Does not cover domestic workers |
| | | | • 2019 is a 'cooling-off period' for the switch from FWCS to SOCSO | • Aims to achieve near equal coverage as citizens | • Migrant workers not covered under the Invalidity Pension Scheme |
| | | | • Contribution rate is the same as citizens: 1.25% of the insured monthly wages; paid by the employer | • Workers to receive support until full recovery, including rehabilitation and disablement benefits | • Workers are not entitled to education loan benefit, vocational training, dialysis treatment or return to work programme (citizen only) |
| | | | | • Dependents to receive benefits | • Exact implementation is yet to be made known |
| | | | | • Onus is on the employer to contribute, but SOCSO claims can be made even if employer fails to contribute | |

* There is no legal provision for SPIKPA. SPIKPA provision is written in policy documents [15].

** Migrant workers were included in SOCSO between 1971 to 1993. In 1996 the Workmen's Compensation Act was amended to specify coverage of migrant workers. In 1998, the FWCS, a private insurance scheme was established to provide social security insurance coverage for migrant works.

***Since January 2019, migrant workers were re-included in SOCSO on near parity terms with citizens.

**Table 3. Major themes and sub-themes related to challenges of healthcare financing and the evolving social security scheme for migrant workers.**

| |
|---|
| **Theme 1: Challenges with healthcare financing for migrant workers** |
| **Weak implementation of health insurance policies** |
| • SPIKPA: Lack of awareness and no insurance card |
| **Institutional discrimination of migrant workers** |
| • Increased non-citizen fees at public healthcare facilities discourages care-seeking |
| • Differential charges for non-citizens perceived to be unfair |
| • Subsidy for foreign workers removed because of the scarce national budget on health |
| **Misalignment between services provided and migrant health needs** |
| • SPIKPA: Inadequate benefits package and coverage of insurance, especially after the increase in non-citizen fees |
| • SPIKPA insurance coverage excludes certain workers and does not cover outpatient treatment |
| **Mistrust in government initiatives by migrant workers and employers** |
| • SPIKPA: Employer uptake unclear |
| **Is the SPIKPA coverage adequate?** |
| **Theme 2: Challenges with the evolving social security scheme for migrant workers** |
| **Weak implementation of social protection policies** |
| • FWCS: Claiming compensation is a complex process |
| • FWCS: Employers deduct medical expenses from compensation |
| **Institutional discrimination of migrant workers** |
| • FWCS: Equality of treatment of workers |
| **Misalignment between services provided and migrant health needs** |
| • FWCS: Inadequate compensation |
| **Mistrust in government initiatives by migrant workers and employers** |
| • FWCS: Employers reluctant to report workplace accidents which affect insurance claims |
| **Potential challenges implementing SOCSO with migrant workers** |
| • SOCSO: 'Too early to tell' |
| • SOCSO: No-fault compensation may lead to excess claims |
| • SOCSO: Uncertainty over the portability of benefits overseas |

## Theme 1: Challenges with healthcare financing for migrant workers

SPIKPA is a mandatory health insurance for migrant workers introduced since January 2011 by the government of Malaysia to alleviate the burden of unpaid bills on the Malaysian public healthcare system, while providing migrant workers with financial risk protection against excessive out-of-pocket healthcare payments. The SPIKPA scheme is mandatory for all documented migrant workers, except domestic and plantation workers, as a necessary pre-requisite for the issuance or renewal of work permits. SPIKPA is a private insurance policy provided by 25 different insurers, with an annual premium of RM120 (USD 30) per migrant worker. The annual coverage of RM10,000 (USD 2,478) was increased to a maximum of RM20,000 (USD 4,956), presumably in keeping with the increase of non-citizen fees at public hospitals. The SPIKPA scheme provides hospitalisation and surgical benefits at public hospitals during employment, while outpatient care, healthcare for pregnancy and attempted suicide or self-harm are excluded. SPIKPA has a 'cashless', e-system, that exempts insured migrant workers from payment of deposits or producing guarantee letters from employers. According to policy wording, insured migrant workers are only required to produce their passport for identity verification at hospital registration counters [15, 34]. (Table 2)

**Weak implementation of health insurance policies.** *SPIKPA: Lack of awareness and no insurance card.* Most interviewed were concerned that SPIKPA, although compulsory, does not provide insurance cards or documents, as such workers and their employers are unaware

of insurance provisions and eligibilities. This is perceived as unfair as workers are forced to pay the annual premiums, but many are reluctant to seek needed care as they were unaware of insurance provisions.

> *"Normally, [when] we have [an] insurance policy, at least we should know [how it works]. We should have documents showing that we have this policy. But in the situation of migrant workers, most of them, they don't have it. They don't have the information [because] they are not given the information. "* CSO-9

Those interviewed shared that by not having a card, workers were unsure whether they were covered by health insurance. This IO interviewee informed of difficulties in claiming insurance, as migrant workers are unsure if they were insured or of their entitlements and the required processes for making a claim.

> *"She got the insurance, [. . .] but when she got hospitalized, there was no proper insurance card for her. She went to the [XX public] hospital. And then, she had to pay deposits up to her discharge, everything she had to pay. The employer didn't [get] involved at all, and she didn't know how to claim. So, everything she had to borrow right and left, to settle the bill, in order to get the check-up."* IO-1

**Institutional discrimination of migrant workers.** *Increased non-citizen fees at public healthcare facilities discourages care-seeking.* Interviewees shared that the removal of healthcare subsidies for non-citizens has resulted in healthcare avoidance among migrant workers, putting population health at risk.

> *"When they removed the subsidy, that was not a very good idea. The migrant thinks it is a high cost. If I don't have enough money, then I won't seek treatment. So, that puts everyone at a higher risk."* CSO-1

This MW interviewee shared that undocumented workers are particularly vulnerable to the increase in medical fees, as they are not covered by the SPIKPA insurance.

> *"They [undocumented workers] just get emergency treatment. Like if the leg is broken or something. They [healthcare workers] just give like temporary treatment, because they cannot afford the bills. Because for the migrant workers, the charges are 200% more expensive. It is very expensive"* MW-1

*Differential charges for non-citizens perceived to be unfair.* One interviewee felt that the increased medical charges were unfair as workers contribute to the Malaysian economy through the payment of the annual levy.

> *"I think the charges should be lowered. Don't discriminate because we are also in Malaysia. For migrant workers, we are not free here. We also have to pay tax to the government with the levy. The levy—one year is RM1,850! We also give contribution to the Malaysia economy. So, why they discriminate [against] us?"* MW-1

While employers are responsible for the payment of levies in policy, in practice levy costs are often deducted from migrant workers' pay.

*Subsidy for foreign workers removed because of the scarce national budget on health*. The steep fee increases for non-citizens were prompted by perceptions that non-citizens took up too much of the healthcare budget. This medical practitioner explained that the ideals of Universal Health Coverage were impractical, in times of financial scarcity.

> *"So, these questions of treating foreign workers and advocating for Universal [Health] Coverage [are] all well respected; but you know, our budget for the MOH is very limited. And the foreign workers only until recently were given free treatment, with a very minimum amount charged. Only recently, [did] the government decide that they [foreign patients] were biting into our budget. So, the direction and directives were given by the MOH that they should be charged [appropriately]."* MD-5

He went on to state that employers should take more responsibility in providing healthcare for workers.

**Misalignment between services provided and migrant health needs.**   *SPIKPA: Inadequate benefits package and coverage of insurance, especially after the increase in non-citizen fees.* Most interviewed felt that the SPIKPA insurance is inadequate in covering costs of medical treatment, especially following the increase in non-citizen fees at public hospitals. This CSO interviewee explained that this may be the reason migrant workers are sent back to home countries after workplace injuries, without receiving adequate medical care in Malaysia.

> *"If you want to give insurance, you [should] give insurance that is in par with Malaysians, you know, Malaysians can get up to RM200,000 to RM500,000 a year. And then we can go to private hospital, it can cover, you know? Public hospitals is almost like free for us already. But, for foreign nationals, it is very high. You know, the cost of giving birth is high, the cost of surgery is high, and hospitalization is very high. But then, the insurance is so low. I think the care for them is not there. They cannot get proper care because the insurance doesn't cover it. So, where we see workers are injured, for example, if they work in a factory and their fingers are cut [. . .] they lose their fingers [and] we see employers sending them back."* CSO-2

According to policy, SPIKPA places the responsibility of healthcare payments upon both the worker and employer. In theory, the worker and employer mutually decide who pays for annual premiums. Both are also responsible for additional hospital charges after medical bills exceed the insurance ceiling, as workers are unable to renew the annual work permit for further employment and employers blacklisted by the Immigration Department thus unable to hire new migrant workers, as a consequence of unpaid hospital bills. In practice, however, workers inevitably bear the burden of paying insurance premiums and excess medical bills.

*SPIKPA insurance coverage excludes certain workers and does not cover outpatient treatment.* Domestic and plantation workers are excluded from mandatory SPIKPA enrolment, with the responsibility of paying for healthcare is placed with the employer. Here, employers could opt to pay directly for healthcare or to enrol these categories of workers into SPIKPA or private insurance schemes. Unfortunately, this lack of standardisation puts workers at the mercy of employers for much needed provision of healthcare.

> *"Having this insurance scheme helps in some ways. But it is very limited, and the amount of insurance coverage is very little. And the worker has to pay for it [pays the premium], except for the plantation and domestic workers [for which] the employer pays. But, other than that, it is the worker who pays for the injuries."* IO-2

As the SPIKPA insurance scheme only provides coverage for hospitalisation and surgery at public hospitals, this CSO interviewee informed that most migrant workers do not utilise the insurance. This maybe because the SPIKPA insurance does not pay for the more commonly sought outpatient treatment at private clinics.

*"It is only when there is an accident or when there is surgery or hospitalization, then you have access, other than that, there is almost totally no care. They lack the awareness and of course, when they go to public hospitals, the fees are quite high, so that kind of discourages them."* CSO-1

As most employers do not pay for the healthcare of workers and migrant workers' pay out-of-pocket for outpatient visits, affordability was a major concern raised.

**Mistrust in government initiatives by migrant workers and employers.** *SPIKPA: Employer uptake unclear*. While both workers and employers are responsible for the payment of hospital bills when the SPIKPA limit is exceeded, employers must ensure that arrears are paid, or risk being blacklisted from recruiting new migrant workers. This industry stakeholder expressed that purchasing health insurance is crucial in protecting employers from unexpected medical bills.

*"I just followed all the requirements by KDN (The Home Ministry); because I don't want [the] company to spend more money for medical, for the foreign worker. That is why I buy the insurance, yeah."* IND-4

Although the SPIKPA insurance scheme is a government policy, it is not governed by law. Some interviewed questioned the enforcement of SPIKPA purchase for migrant workers.

*"In many situations, even though it is part of the conditions by the Malaysian Home Affairs or the government, employers don't buy this insurance, but the work permit is still issued."* IO-1

*Is the SPIKPA coverage adequate*?. We examined data on annual revenues collected by the MOH and outstanding revenues for health services at MOH healthcare facilities under the Fees Act (1951), for citizens and non-citizens from 2008 to 2018. We found that since the 2014 revision of the Fees Act (1951), annual revenues for medical fees collected from non-citizens nearly doubled, from RM 104 million (USD 26 million) in 2014 to RM 182 million (USD 45 million) in 2018, nearing the RM 217 million (USD 54 million) collected from citizens in 2018. Not surprisingly, outstanding revenues for non-citizens tripled during the same period from RM16 million (USD 4 million) in 2014 to RM 50 million (USD 12 million) in 2018 (Fig 2).

While the arrears may have been contributed by undocumented migrants without insurance incurring healthcare payments they cannot afford to pay, these findings in addition to the qualitative evidence of financial barriers to healthcare access, raises questions on the adequacy of the SPIKPA insurance in providing financial risk protection to migrant workers.

Importantly, we were unable to differentiate out-of-pocket contributions of documented migrant workers who had exceeded the SPIKPA threshold, undocumented workers and other migrant populations without health insurance. Limitations of this estimation are mainly due to the aggregated nature of the MOH data used in this analysis. Non-citizens here include documented and undocumented migrant workers, refugees, asylum seekers, expatriates, foreign students, tourists and medical tourists. We propose that a more detailed analysis of individual patient data by these different categories of non-citizens, be conducted in future to examine

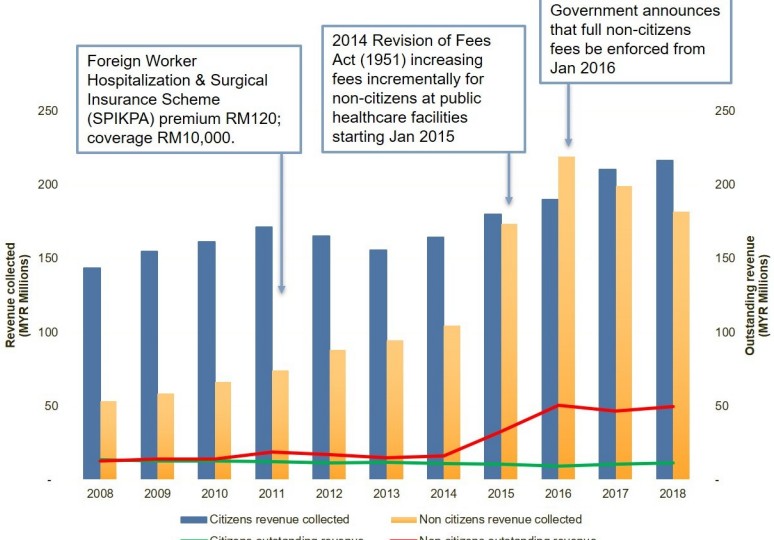

**Fig 2. Revenue collected and outstanding revenue for health services under the Fees Act 1951 by citizenship status, 2008–2018.** Source: Annual Report, Ministry of Health, 2008–2018. [24].

the change of utilisation patterns of health services and expenditure of non-citizens with changes in healthcare policy.

## Theme 2: Challenges with the evolving social security scheme for migrant workers

SOCSO, the national social security scheme in Malaysia is named after the government agency established to provide social security to workers under the Employees' Social Security Act, 1969 (Act 4) [16]. SOCSO which provides insurance to citizens against workplace accidents also covered migrant workers from its establishment in 1971 until 1993. However, since April 1993, migrant workers were exempted from SOCSO and were given protection against occupational disease, and injury and death related to employment under the Workmen's Compensation Act 1952 (Act 273) (WCA). The WCA, a colonial-era legislation enforced by the Department of Labour under the Ministry of Human Resources, was amended in 1996 to provide social security indemnity for migrant workers. Employers were required to insure migrant workers under the FWCS established in 1998. The FWCS is sold by 26 private insurers with annual premiums of RM 72 (USD 18) per migrant worker paid for by employers and providing maximum lump-sum compensations of RM23,000 (USD 5,700) for injuries and RM 25,000 (USD 6,196) for death related to employment, which includes repatriation but not medical expenses [34, 35].

The Malaysian government through the ratification of International Labour Organization (ILO) Equality of Treatment (Accident Compensation) Convention 1925 (No. 19) made an international commitment towards the equality of treatment of citizens and non-citizen workers in terms of accident compensation for occupational injury and industrial accidents [36, 37]. Towards this end, SOCSO was reintroduced for migrant workers starting January 2019, offering the Foreign Workers Employment Injury Scheme (EI Scheme) with similar protections as citizens. All documented migrant workers in Malaysia, except domestic workers, are eligible for SOCSO. While migrant workers newly recruited in 2019 would be automatically covered under SOCSO's EI Scheme, existing migrant workers would receive EI coverage upon

the expiry of their FWCS coverage. Thus 2019 is considered a 'cooling-off' period before the full enforcement of SOCSO. The SOCSO contribution rate for migrant workers is the same as citizens, with employers responsible for the payment of the monthly contribution of 1.25% of the insured migrant workers' monthly wages [27, 38].

The EI Scheme provides superior protection for migrant workers compared to FWCS and while there isn't absolute parity with citizens, it is seen as a step forward. Under the EI Scheme, migrant workers would receive medical, temporary and permanent disablement, dependants, funeral, and rehabilitation benefits, while the education loan benefits, dialysis treatment, vocational and return-to-work programmes are restricted to citizens. Migrant workers also will not be covered under SOCSO's Invalidity Pension Scheme [39].

Under the EI scheme, workers with permanent disablements are entitled to periodic payments of up to 90 percent of the average workers' wage, which is substantially higher than the maximum lump-sum compensation of RM 23,000 (USD 5,700) offered by the FWCS. In the instance of a workplace injury or occupational disease, medical expenses at public clinics and hospitals would be borne by SOCSO, which is unlike provisions under the FWCS, where employers would first pay for treatment and later be reimbursed from the compensation amount. The EI scheme also provides temporary disablement benefits of 80 percent of the average workers' wage for injured workers certified 'unfit' by a medical officer for at least 4 days, not including the day of the accident [36, 38]. (Table 2)

**Weak implementation of social protection policies.** *FWCS*: *Claiming compensation is a complex process*. Interviewees explained that claiming compensation under the FWCS was a lengthy and administratively complex process. The WCA does not provide detailed guidelines on the procedures necessary for claiming compensation leading to much confusion. Also, migrant workers could only claim compensation after they fully recover and have a medical report detailing their injuries. This CSO interviewee shared that injured workers were forced to support themselves financially during the recovery period, even though they were unable to work.

> *"That process takes us, four months to six months, around that [compensation claims]. At that time, he's not getting a salary. He was staying, just lock himself up in the room, in a rented place. And he has to do dressing and everything."* CSO-3

*FWCS*: *Employers deduct medical expenses from compensation*. With the FWCS, employers are entitled by law to deduct their portion of advanced medical payments from the compensation amount. As explained by this IO interviewee, the injured worker may finally receive very little compensation after deducting employer's expenses.

> *"When there is an accident, the cost of medication is very high. So, when they receive the compensation, and then their employer will deduct for the medical expenses that he has borne in advance. You see the worker goes back with nothing. He may have lost his arm or his leg, but he doesn't get anything else from that."* IO-2

**Institutional discrimination of migrant workers.** *FWCS*: *Equality of treatment of workers*. Concerns were raised regarding the equality of treatment of migrant workers in Malaysia, as until recently different social security policies covered citizen and non- citizen workers. This IO interviewee informed that the Workmen's Compensation Act is not in compliance to the ILO's norms of equality of treatment.

> *"Interestingly, Malaysia is a signatory to the [ILO] Convention 19. Basically, what Convention 19 says is that Malaysia is committed to having a non-discriminatory policy in terms of*

*workplace accidents. So, what interesting about Malaysia is that we have two policies, two systems of distributing compensation. One is the SOCSO compensation [for Malaysians]; they look at it in a broader way, they also look it in to the long-term recovery. Whereas under the Workmen Compensation [FWCS] it is one time, lump-sum payment. It [FWCS is] supposed to be faster, but the [compensation] amount is very little for accidents. The amount of pay-out is not very substantial, it is too little [. . .]. And that is one of the concerns that have been raised." IO-2*

She explained that migrant workers received lump-sum payments for workplace injuries. While this was supposed to be administratively simpler for workers returning to their home countries, compensation amounts were inadequate.

**Misalignment between services provided and migrant health needs.**   *FWCS*: *Inadequate compensation*. Many interviewed complained about the meagre FWCS compensation pay-outs that were considered as insufficient to reimburse the immediate medical costs of managing acute injuries, not to mention injuries requiring long-term care or rehabilitation. This CSO interviewee explained that to avoid excessive medical bills, some employers prefer to send their workers back to home countries without claiming compensation.

*"If there is an injury, the company has to pay the bill first, and then later only, they can claim from the insurance company. So, there is a limit. It is not enough to cover like serious injuries or long term [treatment]. So, after that money finishes and the company just releases the worker." CSO-1*

Those interviewed also informed that compensation was grossly insufficient in cases of death, where the cost of repatriation of deceased workers' bodies would inevitably exhaust the entire compensation amount.

*" For us, this is very much underinsured. Because sometimes, for example, if the worker is from East Indonesia, Timor Leste, those areas [remote regions in Indonesia]. The charges can be as high as RM18,000, just to send the deceased body back. When we deduct all these, very little is left for the deceased family, and this shouldn't happen." IO-1*

Compensation amounts were considered especially unjust in case of death and permanent disability, where lump-sum payments are incomparable to the loss of a lifetime's earnings.

**Mistrust in government initiatives by migrant workers and employers.**   *FWCS*: *Employers reluctant to report workplace accidents which affect insurance claims*. Compensation claims for FWCS could only be done through the employer. Employers must initiate the process by reporting workplace accidents resulting in disablement or death to the Department of Labour within ten days of the accident. Interviewees explained that some employers were reluctant to report workplace accidents to avoid investigations into occupational safety practices. This may be due to the perception that insurance claims were linked to occupational health safety enquiries, as the Department of Safety of Health (DOSH) is also part of the Department of Labour.

*"Because the insurance claim is also tied to the DOSH. If the employer makes a report to the Labour Department, DOSH will come into the picture. Suppose there is negligence in the part of the employer, the employer will be fined." CSO-2*

There were various reasons why employers would rather send their workers home after occupational injuries, rather than initiating compensation claims. While compensation is

notoriously inadequate, employers would also prefer not to be investigated on workplace safety standards. Others would attempt to preserve industrial reputation, as well as avoid the inevitable increase in insurance premiums after claims are made.

> *"He lost his hand. It was the fault of the machine. He has to press a button to hold on to the stamping. So, he put his hand in the stamp [and accidently] stamped it. So, he lost his whole hand. And he's the fourth person in that company. So, what normally [happens is], the employer will send them off [back home]"* CSO-3

In this case, this CSO interviewee explained that it would be cheaper to send the migrant worker back to their home country, rather than to conduct a proper enquiry into workplace injuries or change faulty equipment.

**Potential challenges implementing SOCSO with migrant workers.**   *SOCSO*: *'Too early to tell'*. While there was a general positive feeling of anticipation for the inclusion of migrant workers into SOCSO, many of those interviewed felt it was too early to comment on the benefits or difficulties of SOCSO for migrant workers. This participant explained that SOCSO could only be properly evaluated after there is a 'test case' or an accident that can be claimed.

> *"By bringing it back in [SOCSO for migrant workers], the problem we face is that it is difficult to implement [and] to administer. But we won't see it until there is an accident, [. . .] an accident that can be claimed. If there is no accident, then it is just an insurance. You can buy insurance, you won't see anything until you get into an accident—whether the insurance is good or not."* AC-3

*SOCSO*: *No-fault compensation may lead to excess claims*. With the 'no-fault compensation' migrant workers are not required to prove any negligence on the part of the employer to receive compensation. Although both FWCS and SOCSO offer 'no-fault compensation', this academic interviewee made clear that unlike FWCS with its small lump-sum payment, SOCSO has significantly larger compensation amounts, increasing the potential for abuse of SOCSO's provision of 'no-fault compensation'.

> *"The idea of 'no-fault' compensation is if you cut your finger in the workplace; I don't care whether you cut it purposefully or cut it accidentally. [Or if] it was because your employer didn't provide the care, or it was you who didn't work according to what you are supposed to do. So, it doesn't matter! [Social Security] will pay you."* AC-3

*SOCSO*: *Uncertainty over the portability of benefits overseas*. Unlike FWCS which has lump-sum disbursement of funds, the SOCSO scheme provides for a mix of lump-sum and periodic disbursements. A concern raised by interviewees was on the portability of benefits, where administrative difficulties in compensating of migrant workers or next-of-kin in home countries are anticipated.

> *"One of the difficulties that they [may] find is in terms of portability and the issue of administration. Sometimes, the migrant worker has got a very permanent [or]serious injuries, he may opt to go back. So, how do you continue compensating him and going for rehabilitation and things like that. "* IO-2

Compensation of returnees or family members in home countries would require close cooperation and coordination between migrant-sending and receiving countries, including

mechanisms for identifying and contacting recipients and transferring cash payments. Another issue raised was the availability and adequacy of healthcare in home countries to deliver medical or rehabilitative treatment promised by the social security scheme.

## Discussion

To date a security lens has been applied to migration policy in Malaysia [18], however, this may be changing with the imperative towards achieving the United Nation's Sustainable Development Goals [40]. Although government policy requiring health workers to report undocumented migrants creates a climate of fear around care-seeking among non-citizens [14], the inclusion of migrant workers into the national social security scheme, SOCSO, is a promising step towards realising parity of benefits with citizens [41].

Citizens' concerns around resource constraints is a popular narrative in many countries globally, with common threads of blaming non-citizens for diminishing public resources, like healthcare, education and housing [42–45]. The majority of Malaysians surveyed in a 2019 study, believed that migrant workers should not receive the same pay or benefits as locals [46]. Negative public perceptions of migrants are reinforced by discriminatory policies, which imply that migrants are not worthy of the same welfare provisions as citizens. Public attitudes also play a role in shaping policy and create difficulty in reallocating domestic resources towards migrant populations, even when the solutions are shown to be more cost-effective. A European cost analysis study found significant cost savings through timely treatments for irregular migrants and uninsured EU citizens in a primary care setting. Timely treatment in cheaper primary care compared to more expensive hospital care was estimated to save between 49 to 100 percent of direct medical and non-medical costs of hospitalisations [47].

The economic contributions of migrant workers in Malaysia though significant, are not easily quantifiable. Beyond alleviating labour shortages and increasing productivity, migrant workers, like all consumers in Malaysia, pay a consumption tax or the Sales and Services Tax (SST), a regressive form of indirect taxation. As is the case in Singapore, migrant workers and their employers also contribute in terms of annual levy payments, which may be considered a form of labour tax [7, 48], not earmarked towards the workers' benefit. In Malaysia, these levies generate revenue of close to RM 3 billion (USD 710 million) annually (S2 Table. Estimated annual levies collected for migrant workers by sector, 2019). The annual levy is one of many government fees collected for the employment of migrant workers, which also includes a security bond which varies by nationality [RM 250 to RM 1,500 (USD 62-USD 372)] [49].

We found that both migrant workers and employers lack awareness of SPIKPA enrolment and entitlements, which may explain its limited uptake. The claim-loss ratios for SPIKPA are reported to be extremely low, with the pay-outs approximating 10 percent of premium revenues in 2015 [50]. From the insurance providers' perspective, profits are maximized when claims rates are low. Therefore, it is not in the insurance providers' interest for migrant workers to be aware of their SPIKPA entitlements and claim accordingly.

This paper questions the adequacy of SPIKPA in providing financial risk protection for migrant workers, given the high user fees at public hospitals. While the non-citizen user charges being recouped have increased, unpaid revenues have increased in parallel, raising the questions of adequacy of insurance coverage. We estimate that revenues of RM 191 million (USD 47 million) in SPIKPA premiums were collected by private insurers from 1.6 million workers in 2019 alone, a staggering sum almost four times the amount of outstanding revenue in medical fees incurred by non-citizens in government health facilities. We suggest that the government evaluates the insurance provision for non-citizens and considers combining all contributions into a common pool under government oversight. Specifically, a study should

be conducted to examine the feasibility of insuring other non-citizen populations like refugees, asylum seekers and undocumented workers together with documented migrant workers covered by SPIKPA. REMEDI, an innovative medical insurance scheme for refugees launched by the United Nations High Commissioner for Refugees (UNHCR), Malaysia [22, 51], is currently suspended by the private insurer as loss-making due to high claim rates and poor enrolment [52]. Having a larger pool of enrolees in a comprehensive, government-controlled insurance scheme could present significant cost savings for migrants and would help the MOH recoup unpaid bills by non-citizens. Further research should also investigate the extent and scope of out-of-pocket payments for healthcare among migrant workers, in order to inform where SPIKPA insurance provisions may be best targeted.

Undocumented workers and domestic workers are excluded from health and social security policies, while plantation workers remain excluded from health policies. These groups remain vulnerable to exploitation and financially catastrophic healthcare expenses. While undocumented workers are often excluded from public health insurance schemes despite the cost-savings of enrolment [47, 53, 54]. In Thailand, undocumented workers can enrol in a MOH-backed dedicated migrant health insurance, although challenges remain [55]. The levy contributions, if redirected to the MOH, could function as a funding source towards insuring all migrant populations, including the previously excluded domestic, plantation and undocumented workers, as well as refugees and asylum seekers. In addition, we urge the government to consider providing a more comprehensive coverage of outpatient, inpatient and rehabilitative services across the entire spectrum of healthcare.

SOCSO offers higher level protection for workplace accidents compared to FWCS [39], however, its implementation remains unclear. One issue not communicated is the potential overlap with SPIKPA, as medical expenses for migrant workers' related workplace injuries may now be provided under SOCSO. Issues of portability of social security benefits between migrant sending and receiving countries could be improved through regional and bilateral partnerships, such as Memorandums of Understanding to enhance referral mechanisms to ensure the proper management of returnees [56].

Deficiencies in domestic legislation apply to Malaysians as well as migrant workers, such as the lack of health entitlements for workers under the Employment Act [57]. We suggest that the Employment Act be amended to specify the employer's responsibility for the provision of healthcare for all workers. This revision will benefit everyone, bringing Malaysia in line with the SDGs and the concept of equality of benefits.

Currently, migrant workers are regulated through immigration laws enforced by the Ministry of Home Affairs (MOHA) and labour laws enforced by the Ministry of Human Resources (MOHR), with health and welfare seemingly of secondary concern. Importantly, considering the recent re-emergence of polio among migrant populations in the Eastern Malaysian state of Sabah, health should be at the forefront of migration policy [58]. The government should consider establishing a cabinet-level 'Migrant Working Group' with representatives from each Ministry, to facilitate discussion and movement towards a 'Health in All' policies approach.

This study has some limitations. We may have incurred selection bias by sampling known participants during the initial purposive sampling of attendees of a migrant health stakeholder workshop. Nevertheless, we were able to mitigate this by subsequent snow-ball sampling and contacting stakeholders via LinkedIn. We were mindful that participants may have been providing socially acceptable responses particularly towards sensitive questions, thus we were careful to ask open-ended questions in a non-confrontational manner, and triangulated findings by interviewing different stakeholders and document review. While the qualitative nature of this study prevents generalisation of findings, we were able to gain perspective of 'real world' challenges faced by migrant workers with health financing and social security schemes

through the experience of diverse stakeholders, including migrant workers and their representatives, employers and health professionals. Also, we acknowledge the lack of available data on migrant health insurance coverage, SPIKPA uptake and utilisation, which would have been useful in this policy analysis. In this paper, we attempt to use publicly available quantitative data to contextualise findings.

This policy analysis is unique as it combines qualitative interviews with document review, with contextual quantitative findings, to examine the adequacy of available healthcare financing and social security schemes for migrant workers in Malaysia. We have suggested multi-stakeholder policy interventions both in Malaysia and in migrant-sending countries.

## Conclusion

Migrant health policy in Malaysia, like many other countries worldwide, embodies the conflict between state sovereignty, healthcare rationing and international commitments towards maintaining health and social security for the entire population, including migrant workers. Malaysia is moving towards a more inclusive approach for improved population health, with the provision of SOCSO for documented migrant workers, but more needs to be done. Here we suggest the expansion of the SPIKPA insurance scheme to include all migrant populations in Malaysia and broadening of its scope towards more comprehensive coverage, including essential primary care services.

## Supporting information

**S1 File. Interview guide.**
(DOCX)

**S1 Table. Charges for Malaysian citizens and non-citizens at public clinics and hospitals.**
(DOCX)

**S2 Table. Estimated annual levies collected for migrant workers by sector, 2019.**
(DOCX)

## Author Contributions

**Conceptualization:** Tharani Loganathan, Nicola S. Pocock.

**Data curation:** Tharani Loganathan, Zhie X. Chan.

**Formal analysis:** Tharani Loganathan, Zhie X. Chan.

**Funding acquisition:** Tharani Loganathan, Nicola S. Pocock.

**Methodology:** Tharani Loganathan.

**Project administration:** Tharani Loganathan, Zhie X. Chan.

**Software:** Tharani Loganathan, Zhie X. Chan.

**Supervision:** Nicola S. Pocock.

**Validation:** Zhie X. Chan, Nicola S. Pocock.

**Visualization:** Tharani Loganathan, Nicola S. Pocock.

**Writing – original draft:** Tharani Loganathan.

**Writing – review & editing:** Tharani Loganathan, Zhie X. Chan, Nicola S. Pocock.

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
