## [Decision Letter · Decision Letter 0]

28 Aug 2020

PONE-D-20-11263

Healthcare financing and social protection policies for migrant workers in Malaysia

PLOS ONE

Dear Dr. Loganathan,

Thank you for submitting your manuscript to PLOS ONE. After careful consideration, we feel that it has merit but does not fully meet PLOS ONE’s publication criteria as it currently stands. 

The manuscript has been evaluated by four reviewers, and their comments are available below. The reviewers have raised a number of concerns regarding methodological aspects of the study, as well as the interpretation and discussion of your findings. Could you please revise the manuscript to carefully address the concerns raised?

We look forward to receiving your revised manuscript.

Kind regards,

Dario Ummarino, Ph.D.

Associate Editor

PLOS ONE

Journal Requirements:

2. Please ensure that you have provided sufficient details that others could replicate the analyses. For instance, if you developed a questionnaire as part of this study and it is not under a copyright more restrictive than CC-BY, please include a copy, in both the original language and English, as Supporting Information.

Reviewers' comments:

Reviewer's Responses to Questions

**Comments to the Author**

1. Is the manuscript technically sound, and do the data support the conclusions?

Reviewer #1: Partly

Reviewer #2: Yes

Reviewer #3: Partly

Reviewer #4: Yes

2. Has the statistical analysis been performed appropriately and rigorously? 

Reviewer #1: Yes

Reviewer #2: Yes

Reviewer #3: N/A

Reviewer #4: I Don't Know

3. Have the authors made all data underlying the findings in their manuscript fully available?

Reviewer #1: Yes

Reviewer #2: Yes

Reviewer #3: No

Reviewer #4: Yes

4. Is the manuscript presented in an intelligible fashion and written in standard English?

Reviewer #1: Yes

Reviewer #2: Yes

Reviewer #3: Yes

Reviewer #4: Yes

5. Review Comments to the Author

Reviewer #1: Overall assessment:

The article is timely in its contributions to the goals of UHC in an upper-middle-income country that attracts large numbers if migrant labor. It is well-written and the abstract is adequately summarized.

Methodology:

1. Largely qualitative with limited quantitative data; the approach is adequate to answer the objectives of the study.

2. Please clarify Table 1: the total number of participants is said to be 37 (Line 117) while the table gives a total of 44.

Results

1. Minor grammatical issues: line 180 and 181- Use ‘First’ instead of ‘Firstly’ and ‘Second’ instead of ‘Secondly’

2. Further explanation required on line 188. See side comments of main document

3. Correct ‘seven’ instead of 7

4. In general, the results section is detailed enough in terms its description of health policy evolution for migrant labor. However, I am reluctant to agree that the themes accurately address the aim of the study to “evaluate healthcare financing and social security policies for migrant workers; to identify policy gaps and opportunities for intervention” ....

5. …From my reading of the manuscript, the following key themes are evident and should be highlighted and adequately discussed with suggestion for way forward including health financing reforms:

a. Weak implementation of health insurance and social protection policies

-this is evident in the lack of awareness of insurance scope by employer and employee; reluctance to report work-place injuries; a disempowered migrant labor-force that cannot assert itself to exercise its health insurance entitlements

-the authors need also to let readers understand why this (weakness in policy implementation) is happening.

b. Institutional discrimination of migrant workers

-evident in terms of differences in health services coverage between citizens and migrants; differences in compensation in cases of injury

-overall inequities in fees charged, OP and IP care at public facilities; etc.

c. Misalignment between services provided and migrant health needs: (Lines 253 – 4)

-this needs to be discussed in more details including examples and the reasons this is happening and the implications on migrant health

d. Mistrust in government initiatives by migrant workers (as highlighted in lines 467 – 477)

It will help the article a great deal if the authors could feature these major themes more prominently and discuss the sub-themes already in the article, including taking a closer look at their meaning for the goals of UHC in Malaysia (with examples from the interviews).

I’d encourage the authors to aggregate into major themes the sub-themes they have itemized on the article to give it the flow and concreteness it deserves because this article addresses the subject of migrant health which is a major source of inequity in health services.

A suggestion for further research would be a quantitative analysis of out-of-pocket spending by migrant workers; the scope of such spending; the drivers and implications on migrant worker well-being.

Reviewer #2: This study is important under the background of SDGs, especially migrant workers usually neglected by countries. The evidence offered in this study not only helpful in improving corresponding policies in Malaysia, but also give a good reference to other countries. The mixed method used in this study ensured the complementary evidence, and related study design and data analysis are rational. A weakness of this study is that this paper is too long. It’s better if the author could compact this paper. Maybe put some tables into supplementary. Another weakness is that journal article consist of little part of the reference. It makes me doubt that if the author conduced a sufficient review.

Reviewer #3: The paper seeks to “examine the evolution of healthcare policy for migrant workers in Malaysia, while evaluating healthcare financing and social security policies and other options for fiscal space, in order to identify gaps and opportunities to improve migrant health financing and coverage.”

The paper is an important and interest read, researching an understudied population (migrant workers) and is of high policy relevance in Malaysia and beyond, in particular in other context with larger labour migrant populations. The analysis is supported by highly informative tables (in particular Table 3). The context is well-described for a reader who is not very familiar with the Malaysian context.

While this is of great interest, there are some important concerns.

Major comments:

1) This analysis builds upon mixed methods (page 6), combining qualitative analysis (Policy document review, qualitative interviews) and quantitative data. While this is of interest, the abstract and methods section do not provide sufficient description of why was this approach taken and further details about the different methods used:

The policy literature review needs further detail. The authors describe: “For the document review, Malaysian health and labour laws, policy documents, guidelines and circulars relating to healthcare delivery to migrants, and reports of local and international organisations concerning migrant health were retrieved and analysed.”. The following questions remain: How were these identified? Which time frame (the results section write past two decades)? How were they analysed? In my view, it seems like a policy mapping more than review has been carried out (identification of policy and main content).

Quant. Component: “For the quantitative component, we analysed published data on medical revenues collected and outstanding revenues…” Please add additional information about how this information was collected (it sounds like data extraction more than data collection).

Related to these weaknesses, I do think the statement in the Discussion is an exaggeration “This policy analysis is unique as it combines qualitative interviews with document review and examination of the economic evidence, to examine the adequacy of available healthcare financing and social security schemes for migrant workers in Malaysia.” While the qual. analysis is indepth, it seems as the policy review is a policy mapping, and that the quant. data analysis seems to be data extraction/presentation of data. I am not convinced that the quant. data are quant. data, that are analysis as it seems that they are data presented to support information about the context more than analysis. The revised draft should be clearer about when primary data were collected/or analysed. I think it adds value, but should not be presented as quant. analysis unless this is done.

2) Results

o Reporting from qualitative interviews, findings are reported as “Those interviewed” or “interviewees” (page 17). It would be relevant to know more about who said this, given the heterogeneous sample.

Minor comments:

3) Page 4, line 57-58:You refer to the Migration compact. While this is relevant, organisations such as ILO and IOM have been working to improve the rights of migrants for a longer period.

4) Further discussion of Malaysian changes in health policies in relation to trends in the geographical region (or other labour-migrant dependent economies) would be of great interest.

Reviewer #4: The manuscript was well written, easy to read, clear and has minimal typographical or grammatical error.

The following are my suggestions for improvement.

Line 62, 63: Need reference

Line 117: 37 in-depth interviews of 44 individuals: Need to elaborate the division of the 44 individuals to get the 37 in-depth interviews.

Line 172, and 346: Fig 1 and Fig 2. Is this a short form for figure? If yes it is not stated clearly. I cannot find fig 1 nor fig 2, what follow is more of a description

Line 233 and line 405: Qualitative 1 and Qualitative 1.

The presentation of results needs some organisation to make it easier to read and understand.

Example after the topic/heading i.e qualitative 1 (line233) it will be followed by sub-heading (line 234) this is not the case of line 307, there is no sub-heading.

6. PLOS authors have the option to publish the peer review history of their article (what does this mean?). If published, this will include your full peer review and any attached files.

Reviewer #1: **Yes: **Dr Vincent Okungu

Reviewer #2: **Yes: **shanquan chen

Reviewer #3: No

Reviewer #4: No

---

## [Author Response · Author response to Decision Letter 0]

19 Sep 2020

Response to reviewers

Reviewer #1: 

Methodology

1 Please clarify Table 1: the total number of participants is said to be 37 (Line 117) while the table gives a total of 44.

We conducted 37 in-depth interviews, with a total number of 44 participants interviewed.

We aimed to conduct in-depth individual interviews for this study. However, some organisations sent several representatives (2-3 people) to participate in interviews. For most of these interviews, one main person would respond to questions, while the others would supplement with further detail if necessary. 

Inserted in line 119-121 in the methods section:

We conducted 37 in-depth interviews of 44 individuals including those from civil society and international organisations, trade unions, academia, industry, as well as medical doctors, migrant workers and other policy stakeholders (Table 1). Most interviews were conducted on an individual basis, however several interviews were conducted with small groups of 2 or 3 participants from the same organisation.

Results

1 Minor grammatical issues: line 180 and 181- Use ‘First’ instead of ‘Firstly’ and ‘Second’ instead of ‘Secondly’

The sentences were edited in Line 180 -182 in the results section:

First, non-citizens were framed as a potential national security threat requiring collective action. Second, non-citizens were seen as taking up a large portion of the MOH budget meant for Malaysian citizens

2 Further explanation required on line 188. See side comments of main document. Please explain why this was enforced ahead of time.

The Malaysian government initially planned for a staggered phasing out of subsidised fees for non-citizens, but this abruptly changed with immediate enforcement beginning January 1, 2016. The decision to charge patients who were not Malaysian citizens full fees at public healthcare facilities beginning January 1, 2016 was announced during the 2016 budget presentation on 23rd October 2015 by the Prime Minister. No reasons were given for this change in policy implementation. 

The sentences were edited in Line 187 -188 in the results section:

However, full non-citizen fees were enforced from January 2016, two-years ahead of the initial target of 2018, without explanation on the reasons for the changes in implementation [1, 2].

3 Correct ‘seven’ instead of 7

Corrected line 191.

4 From my reading of the manuscript, the following key themes are evident and should be highlighted and adequately discussed with suggestion for way forward including health financing reforms:

a. Weak implementation of health insurance and social protection policies

this is evident in the lack of awareness of insurance scope by employer and employee; reluctance to report work-place injuries; a disempowered migrant labor-force that cannot assert itself to exercise its health insurance entitlements

the authors need also to let readers understand why this (weakness in policy implementation) is happening.

b. Institutional discrimination of migrant workers

evident in terms of differences in health services coverage between citizens and migrants; differences in compensation in cases of injury

overall inequities in fees charged, OP and IP care at public facilities; etc.

c. Misalignment between services provided and migrant health needs: 

(Lines 253 – 4)

this needs to be discussed in more details including examples and the reasons this is happening and the implications on migrant health

d. Mistrust in government initiatives by migrant workers 

(as highlighted in lines 467 – 477)

It will help the article a great deal if the authors could feature these major themes more prominently and discuss the sub-themes already in the article, including taking a closer look at their meaning for the goals of UHC in Malaysia (with examples from the interviews). I’d encourage the authors to aggregate into major themes the sub-themes they have itemized on the article to give it the flow and concreteness it deserves because this article addresses the subject of migrant health which is a major source of inequity in health services.

Thank you for the suggestion of major themes for the results section. We are aware that to avoid confusion, we need to present our findings in an organised narrative fashion. Our results comprise of multiple components and we need to display results of policy review, qualitative and quantitative analysis for financing policy (SPIKPA) and social security policy (FWCS and SOCSO). As such we organised qualitative findings into the four major themes as suggested but retained the main sectional divisions (Evolution of migrant healthcare and social security policies in Malaysia, Challenges with healthcare financing for migrant workers, Challenges with the evolving social security scheme for migrant workers).

In re-organising the qualitative findings into the 4 major themes suggested by the reviewer, we have added text and quotes to provide more explanation. We also clearly labelled each subtheme with the policy concerned (SPIKPA, FWCS, SOCSO).

We could not include UHC into our results section because this was not directly asked during interviews.

These are some of the additional changes made:

(i) Line 206: 

We describe the major challenges and gaps in healthcare financing (SPIKPA) and social security policies (FWCS and SOCSO) in the next sections.

(ii) Line 337 - 343, Line 348 – 358:

SPIKPA insurance coverage excludes certain workers and does not cover outpatient treatment

Domestic and plantation workers are excluded from mandatory SPIKPA enrolment, with the responsibility of paying for healthcare is placed with the employer. Here, employers could opt to pay directly for healthcare or to enrol these categories of workers into SPIKPA or private insurance schemes. Unfortunately, this lack of standardisation puts workers at the mercy of employers for much needed provision of healthcare.

“Having this insurance scheme helps in some ways. But it is very limited, and the amount of insurance coverage is very little. And the worker has to pay for it [pays the premium], except for the plantation and domestic workers [for which] the employer pays. But, other than that, it is the worker who pays for the injuries." IO-2

As the SPIKPA insurance scheme only provides coverage for hospitalisation and surgery at public hospitals, this interviewee informed that most migrant workers do not utilise the insurance. This maybe because the SPIKPA insurance does not pay for the more commonly sought outpatient treatment at private clinics. 

“It is only when there is an accident or when there is surgery or hospitalization, then you have access, other than that, there is almost totally no care. They lack the awareness and of course, when they go to public hospitals, the fees are quite high, so that kind of discourages them.” CSO-1

As most employers do not pay for the healthcare of workers and migrant workers’ pay out-of-pocket for outpatient visits, affordability was a major concern raised.

(iii) Line 485 – 502

Institutional discrimination of migrant workers

FWCS: Equality of treatment of workers

Concerns were raised regarding the equality of treatment of migrant workers in Malaysia, as until recently different social security policies covered citizen and non- citizen workers. This interviewee informed that the Workmen’s Compensation Act is not in compliance to the ILO’s norms of equality of treatment.

“Interestingly, Malaysia is a signatory to the [ILO] Convention 19. Basically, what Convention 19 says is that Malaysia is committed to having a non-discriminatory policy in terms of workplace accidents. So, what interesting about Malaysia is that we have two policies, two systems of distributing compensation. One is the SOCSO compensation [for Malaysians]; they look at it in a broader way, they also look it in to the long-term recovery. Whereas under the Workmen Compensation [FWCS] it is one time, lump-sum payment. It [FWCS is] supposed to be faster, but the [compensation] amount is very little for accidents. The amount of pay-out is not very substantial, it is too little […]. And that is one of the concerns that have been raised." IO-2

She explained that migrant workers received lump-sum payments for workplace injuries. While this was supposed to be administratively simpler for workers returning to their home countries, compensation amounts were inadequate. 

We have now included a Table of the main themes and sub themes, for clarity, which will be particularly useful for Malaysian policymakers interested in specific findings:

Theme 1: Challenges with healthcare financing for migrant workers

Weak Implementation of health insurance policies 

- SPIKPA: Lack of awareness and no insurance card

Institutional discrimination of migrant workers

- Increased non-citizen fees at public healthcare facilities discourages care-seeking

- Differential charges for non-citizens perceived to be unfair 

- Subsidy for foreign workers removed because of the scarce national budget on health

Misalignment between services provided and migrant health needs

- SPIKPA: Inadequate benefits package and coverage of insurance, especially after the increase in non-citizen fees

- SPIKPA insurance coverage excludes certain workers and does not cover outpatient treatment

Mistrust in government initiatives by migrant workers and employers

- SPIKPA: Employer uptake unclear

- Is the SPIKPA coverage adequate?

Theme 2: Challenges with the evolving social security scheme for migrant workers

Weak implementation of social protection policies

- FWCS: Claiming compensation is a complex process

- FWCS: Employers deduct medical expenses from compensation

Institutional discrimination of migrant workers

- FWCS: Equality of treatment of workers

Misalignment between services provided and migrant health needs

- FWCS: Inadequate compensation

Mistrust in government initiatives by migrant workers and employers

- FWCS: Employers reluctant to report workplace accidents which affect insurance claims

Potential challenges implementing SOCSO with migrant workers 

- SOCSO: ‘Too early to tell’ 

- SOCSO: No-fault compensation may lead to excess claims

- SOCSO: Uncertainty over the portability of benefits overseas

5 A suggestion for further research would be a quantitative analysis of out-of-pocket spending by migrant workers; the scope of such spending; the drivers and implications on migrant worker well-being.

Thank you for the suggestion, which we have included in the Discussion:

“Further research should also investigate the extent and scope of out-of-pocket payments for healthcare among migrant workers, in order to inform where SPIKPA insurance provisions may be best targeted.” 

Reviewer #2: 

1 A weakness of this study is that this paper is too long. It’s better if the author could compact this paper. Maybe put some tables into supplementary. 

We have modified the results section in accordance to the first reviewer’s comments and this has improved the flow of the paper. We are also moving Table 2 (S2 Table) and Table 4 (S3 Table) to the Appendix.

2 Another weakness is that journal article consist of little part of the reference. It makes me doubt that if the author conduced a sufficient review.

We conducted a mixed-methods study, wherein qualitative data was analysed together with quantitative data and policy document review. Unfortunately, there are few journal publications on financial or social protection policies for migrant workers in Malaysia. As such, we searched a wider domain consisting of grey literature, and reviewed legal documents, reports and circulars from the government agencies namely the Ministry of Health, the PERKESO, the Immigration Department, Ministry of Home Affairs and others, and reports by local and international organisations.

 

Reviewer #3: 

Major comments:

1 This analysis builds upon mixed methods (page 6), combining qualitative analysis (Policy document review, qualitative interviews) and quantitative data. While this is of interest, the abstract and methods section do not provide sufficient description of why was this approach taken and further details about the different methods used:

We used a mixed-methods approach in this policy analysis to better understand and evaluate healthcare financing and social security policies for migrant workers. Provision of services to migrant workers in Malaysia, is a subject of contention as stakeholders are hesitant in providing citizenship entitlements to non-citizen populations. Also, several agencies have differing responsibilities in regulating migrant workers without clear ownership of policies concerning health and social security. We felt the mixed methods approach was necessary for this complex subject, to identify and understand gaps in policy and areas for intervention. 

a. The policy literature review needs further detail. The authors describe: “For the document review, Malaysian health and labour laws, policy documents, guidelines and circulars relating to healthcare delivery to migrants, and reports of local and international organisations concerning migrant health were retrieved and analysed.”. The following questions remain: How were these identified? Which time frame (the results section write past two decades)? How were they analysed? In my view, it seems like a policy mapping more than review has been carried out (identification of policy and main content).

We have clarified in Methods (Line 110 – 133) where policy documents were retrieved from and the time period:

“For the document review, Malaysian health and labour laws, policy documents, guidelines and circulars relating to healthcare delivery to migrants, and reports of local and international organisations concerning migrant health from January 2001 to July 2019 were retrieved from relevant Ministry and local organisation websites (MOH, Ministry of Human Resources, Department of Statistics, PERKESO) and international organisation websites (IOM, ILO, WHO) and reviewed.”

We did not conduct policy mapping and clarify here that documents were reviewed. We reviewed policies to assess how they have evolved over time (Figure 1) and in relation to our in-depth qualitative findings, as elaborated in the Results.

b. Quant. Component: “For the quantitative component, we analysed published data on medical revenues collected and outstanding revenues…” Please add additional information about how this information was collected (it sounds like data extraction more than data collection).

We descriptively analysed publicly available secondary data, and did no primary survey work. We have clarified in Methods (Line 146 - 147):

“To contextualise the policy review and qualitative findings, we analysed publicly available secondary data on medical revenues collected and outstanding revenues in accordance with the Fees Act 1951…This descriptive analysis aimed to examine changes in revenues…”

We also clarified the methods in the Abstract, Line 30-32:

“A mixed-methods analysis combining major themes from qualitative interviews with policy document reviews was conducted. Descriptive analysis of publicly available secondary data, namely revenues collected at government healthcare facilities, was conducted to contextualise the policy review and qualitative findings.”

c. Related to these weaknesses, I do think the statement in the Discussion is an exaggeration “This policy analysis is unique as it combines qualitative interviews with document review and examination of the economic evidence, to examine the adequacy of available healthcare financing and social security schemes for migrant workers in Malaysia.” While the qual. analysis is indepth, it seems as the policy review is a policy mapping, and that the quant. data analysis seems to be data extraction/presentation of data. I am not convinced that the quant. data are quant. data, that are analysis as it seems that they are data presented to support information about the context more than analysis. The revised draft should be clearer about when primary data were collected/or analysed. I think it adds value, but should not be presented as quant. analysis unless this is done.

As above, we have clarified that the descriptive quantitative analysis was used to contextualise the policy and qualitative findings. We have revised the following in the Discussion (Line 708 -709): 

“This policy analysis is unique as it combines qualitative interviews with document review, with contextual quantitative findings, to examine the adequacy of available healthcare financing and social security schemes for migrant workers in Malaysia.”

We experienced difficulty sourcing data from the Ministry of Health, Malaysia on this subject and have acknowledged this lack of data as a study limitation. (Line 704 -707)

“Also we acknowledge the lack of available data on migrant health insurance coverage, SPIKPA uptake and utilisation, which would have been useful in this policy analysis. In this paper, we attempt to use publicly available quantitative data to contextualise findings.”

2 Results

Reporting from qualitative interviews, findings are reported as “Those interviewed” or “interviewees” (page 17). It would be relevant to know more about who said this, given the heterogeneous sample.

We have clarified ‘This CSO interviewee’ on page 17 and have made changes throughout the Results section indicating which type of informant is quoted.

Minor comments:

3 Page 4, line 57-58: You refer to the Migration compact. While this is relevant, organisations such as ILO and IOM have been working to improve the rights of migrants for a longer period.

Thank you for the comment. We agree but don’t see it as necessary to mention here. We have made the following edits in Line 58-59:

International commitment towards protecting migrant workers’ rights is most recently embodied in the 2018 Global Compact for Safe, Orderly and Regular Migration.

4 Further discussion of Malaysian changes in health policies in relation to trends in the geographical region (or other labour-migrant dependent economies) would be of great interest.

Thank you for the suggestion. We have revised to mention Singapore in the Discussion as the only other ASEAN receiving country that charges a migrant worker levy. EU countries and Thailand are referred to vis a vis their policies on migrant health and migrant healthcare financing. Given the length of the paper we prefer not to expand on general health policies of surrounding countries in the Discussion (there is a SEA Lancet series and other papers which examine health policies/UHC in ASEAN countries available elsewhere).

Discussion[3], Line 631- 635

As is the case in Singapore, migrant workers and their employers also contribute in terms of annual levy payments, which may be considered a form of labour tax [4, 5], not earmarked towards the workers’ benefit. In Malaysia, these levies generate revenue of close to RM 3 billion (USD 710 million) annually

Reviewer #4: 

1 Line 62, 63: Need reference

Adding these references:

1. Kanapathy V. Migrant workers in Malaysia: An overview. Country paper prepared for the Workshop on an East Asian Cooperation Framework for Migrant Labour, Kuala Lumpur: 2006.

2. ILO Regional Office for Asia and the Pacific. Review of labour migration policy in Malaysia. Tripartite Action to Enhance the Contribution of Labour Migration to Growth and Development in ASEAN (TRIANGLE II Project) Bangkok: ILO, 2016.

2 Line 117: 37 in-depth interviews of 44 individuals: Need to elaborate the division of the 44 individuals to get the 37 in-depth interviews.

We conducted 37 in-depth interviews, with a total number of 44 participants interviewed.

Inserted in line 119-121 in the methods section:

We conducted 37 in-depth interviews of 44 individuals including those from civil society and international organisations, trade unions, academia, industry, as well as medical doctors, migrant workers and other policy stakeholders (Table 1). Most interviews were conducted on an individual basis, however several interviews were conducted with small groups of 2 or 3 participants from the same organisation.

3 Line 172, and 346: Fig 1 and Fig 2. Is this a short form for figure? If yes it is not stated clearly. I cannot find fig 1 nor fig 2, what follow is more of a description

We follow PLOS naming guides for figures (Fig1 etc.). Figures are uploaded separately into the submission system and can be found at the end of the manuscript.

4 Line 233 and line 405: Qualitative 1 and Qualitative 1.

The presentation of results needs some organisation to make it easier to read and understand.

Example after the topic/heading i.e qualitative 1 (line233) it will be followed by sub-heading (line 234) this is not the case of line 307, there is no sub-heading.

We have modified the results section in accordance to the first reviewer’s comments and we feel this has improved the flow of the paper. 

We have removed headings that label themes e.g. Qualitative 1 and Qualitative 2. Headings and subheadings have been changed accordingly.

 

References

1. Ministry of Health Malaysia. Circular of the Secretary General, Ministry of Health, Malaysia no. 8/2015: Implementation of full charges Fees Order (Medical) (Service cost) 2014. 2015.

2. Budget 2016: Full speech by PM Najib. The Star. 2015 23rd October 2015.

3. Kanapathy V. Migrant workers in Malaysia: An overview. Country paper prepared for the Workshop on an East Asian Cooperation Framework for Migrant Labour, Kuala Lumpur: 2006.

4. ILO Regional Office for Asia and the Pacific. Review of labour migration policy in Malaysia. Tripartite Action to Enhance the Contribution of Labour Migration to Growth and Development in ASEAN (TRIANGLE II Project) Bangkok: ILO, 2016.

5. Chan Chee Khoon. Re-balancing ASEAN Integration: Medical Tourism vs Migrants’ Health? . Thinking ASEAN 2017 August 2017:15-7.

---

## [Decision Letter · Decision Letter 1]

17 Nov 2020

PONE-D-20-11263R1

Healthcare financing and social protection policies for migrant workers in Malaysia

PLOS ONE

Dear Dr. Loganathan,

Thank you for submitting your revised manuscript to PLOS ONE. After careful consideration, we feel that it has merit but does not fully meet PLOS ONE’s publication criteria as it currently stands. 

The manuscript has been evaluated by four reviewers (three old, one new), and their comments are available below. I was a reviewer of the previous version the manuscript, and are now serving as a Guest Editor for the manuscript. The manuscript has improved substantially and three reviewers were satisfied with your responses and revisions. However, a new reviewer reiterate concerns raised regarding your claims of a mixed-methods study and triangulation is unclear. While you have added clarification on the policy analysis, we ask you to revise the manuscript to carefully address the methods concerns raised. A detailed description in the Study design section of the manuscript of why a mixed-method study design was chosen would be helpful, and reflection on its strengths and limitation in the Discussion section.

We look forward to receiving your revised manuscript.

Kind regards,

Kristine Husøy Onarheim

Guest Academic Editor

PLOS ONE

Reviewers' comments:

Reviewer's Responses to Questions

**Comments to the Author**

1. If the authors have adequately addressed your comments raised in a previous round of review and you feel that this manuscript is now acceptable for publication, you may indicate that here to bypass the “Comments to the Author” section, enter your conflict of interest statement in the “Confidential to Editor” section, and submit your "Accept" recommendation.

Reviewer #1: All comments have been addressed

Reviewer #2: All comments have been addressed

Reviewer #4: All comments have been addressed

Reviewer #5: (No Response)

2. Is the manuscript technically sound, and do the data support the conclusions?

Reviewer #1: Yes

Reviewer #2: Yes

Reviewer #4: Yes

Reviewer #5: Yes

3. Has the statistical analysis been performed appropriately and rigorously? 

Reviewer #1: Yes

Reviewer #2: Yes

Reviewer #4: (No Response)

Reviewer #5: Yes

4. Have the authors made all data underlying the findings in their manuscript fully available?

Reviewer #1: Yes

Reviewer #2: Yes

Reviewer #4: Yes

Reviewer #5: Yes

5. Is the manuscript presented in an intelligible fashion and written in standard English?

Reviewer #1: Yes

Reviewer #2: Yes

Reviewer #4: Yes

Reviewer #5: Yes

6. Review Comments to the Author

Reviewer #1: The authors have made a great effort to reorganize the manuscript and address reviewer concerns. It should be able to be published with minor edits

Reviewer #2: this study is useful, and the authours modeifed the manuscript based on my comments, and no more comments from me.

Reviewer #4: The author has sufficiently address the comments made by reviewer. This is an important paper that may contribute to improvement of migrant livelihood in Malaysia.

Reviewer #5: The previous reviewers questioned whether the way qualitative and quantitative work has been conducted and analyzed is consistent with the claim of a mixed-methods design. The authors' response is still unclear. Since there was no primary quantitative data collection and the methods employ no clear strategy (and clear description) e.g. triangulation to convincingly demonstrate validation of study questions using different methods, the mixed-methods claim is still unconvincing. Authors need to clarify this in methods and demonstrate this in results and discussion.

7. PLOS authors have the option to publish the peer review history of their article (what does this mean?). If published, this will include your full peer review and any attached files.

Reviewer #1: **Yes: **Vincent Okungu

Reviewer #2: No

Reviewer #4: No

Reviewer #5: No

---

## [Author Response · Author response to Decision Letter 1]

21 Nov 2020

We thank the previous three reviewers for taking the time to review and accept changes to the revised version of our manuscript. Their substantive comments and suggestions have strengthened the paper. We address the minor outstanding point below.

Response to the Editor:

The manuscript has been evaluated by four reviewers (three old, one new), and their comments are available below. I was a reviewer of the previous version the manuscript, and are now serving as a Guest Editor for the manuscript. The manuscript has improved substantially and three reviewers were satisfied with your responses and revisions. However, a new reviewer reiterate concerns raised regarding your claims of a mixed-methods study and triangulation is unclear. While you have added clarification on the policy analysis, we ask you to revise the manuscript to carefully address the methods concerns raised. A detailed description in the Study design section of the manuscript of why a mixed-method study design was chosen would be helpful, and reflection on its strengths and limitation in the Discussion section.

Use of the term ‘mixed methods analysis’ appears to be the sticking point. We omit use of this term in the revised Study design section as follows (Page 6, Line 93- 98):

Study design 

Policy document review and thematic analysis of qualitative interviews were combined with secondary quantitative data to evaluate healthcare financing and social security policies for migrant workers. We combined major themes from interview data with policy analysis and descriptive quantitative data is presented to contextualize findings, in an integrated Results section. Our mixed methods approach provides insights at critical moments in the analysis, rather than a standardized mixed methods research approach, which has been critiqued for taking an unreflective and mechanical approach to knowledge production[1].

There are no current references to a mixed methods analysis in the Limitations section. As such, we see no need to make changes here. 

Response to Reviewer 5

Reviewer #5: The previous reviewers questioned whether the way qualitative and quantitative work has been conducted and analyzed is consistent with the claim of a mixed-methods design. The authors' response is still unclear. Since there was no primary quantitative data collection and the methods employ no clear strategy (and clear description) e.g. triangulation to convincingly demonstrate validation of study questions using different methods, the mixed-methods claim is still unconvincing. Authors need to clarify this in methods and demonstrate this in results and discussion.

Use of the term ‘mixed methods analysis’ appears to be the sticking point. We omit use of this term in the revised Study design section, and clearly state that methods were combined in an integrated Results section. We specify that secondary quantitative data were descriptively analysed (Page 6, Line 93- 98):

Study design 

Policy document review and thematic analysis of qualitative interviews were combined with secondary quantitative data to evaluate healthcare financing and social security policies for migrant workers. We combined major themes from interview data with policy analysis and descriptive quantitative data is presented to contextualize findings, in an integrated Results section. Our mixed methods approach provides insights at critical moments in the analysis, rather than a standardized mixed methods research approach, which has been critiqued for taking an unreflective and mechanical approach to knowledge production[1]

References

1. Timans R, Wouters P, Heilbron J. Mixed methods research: what it is and what it could be. Theory and Society. 2019;48(2):193-216. doi: 10.1007/s11186-019-09345-5.

---

## [Editor Report · Decision Letter 2]

25 Nov 2020

Healthcare financing and social protection policies for migrant workers in Malaysia

PONE-D-20-11263R2

Dear Dr. Loganathan, 

We’re pleased to inform you that your manuscript has been judged scientifically suitable for publication and will be formally accepted for publication once it meets all outstanding technical requirements. 

Kind regards,

Kristine Husøy Onarheim

Guest Editor

PLOS ONE

---

## [Editor Report · Acceptance letter]

27 Nov 2020

PONE-D-20-11263R2 

Healthcare financing and social protection policies for migrant workers in Malaysia 

Dear Dr. Loganathan:

I'm pleased to inform you that your manuscript has been deemed suitable for publication in PLOS ONE. Congratulations! Your manuscript is now with our production department. 

Kind regards, 

on behalf of

Dr. Kristine Husøy Onarheim 

Guest Editor

PLOS ONE